# Sleep and Chronobiology as a Key to Understand Cluster Headache

Laura Pilati [1,2,†], Angelo Torrente [1,*,†], Paolo Alonge [1], Lavinia Vassallo [1], Simona Maccora [1,3], Andrea Gagliardo [1,4], Antonia Pignolo [1,5], Salvatore Iacono [1], Salvatore Ferlisi [1,6], Vincenzo Di Stefano [1], Cecilia Camarda [1] and Filippo Brighina [1]

1   Department of Biomedicine, Neurosciences and Advanced Diagnostics (Bi.N.D.), University of Palermo, 90127 Palermo, Italy
2   Headache Center "Casa della Salute Cittadella San Rocco", AUSL Ferrara, 44121 Ferrara, Italy
3   Neurology Unit, ARNAS Civico di Cristina and Benfratelli Hospitals, 90127 Palermo, Italy
4   Clinical Neurophysiology Unit, Sleep Lab, "Clinical Course", 90143 Palermo, Italy
5   Department of Neuroscience, "S. Giovanni di Dio" Hospital, 88900 Crotone, Italy
6   Azienda Sanitaria Locale di Collegno e Pinerolo, 10093 Collegno, Italy
*   Correspondence: angelo.torrente@unipa.it; Tel.: +39-091-655-4780; Fax: +39-091-655-2974
†   These authors contributed equally to this work.

**Abstract:** The cluster headache is a primary headache characterized by attacks of unilateral pain associated with ipsilateral cranial autonomic features. These attacks recur in clusters during the years alternating with periods of complete remission, and their onset is often during the night. This annual and nocturnal periodicity hides a strong and mysterious link among CH, sleep, chronobiology and circadian rhythm. Behind this relationship, there may be the influence of genetic components or of anatomical structures such as the hypothalamus, which are both involved in regulating the biological clock and contributing even to the periodicity of cluster headaches. The bidirectional relationship manifests itself also with the presence of sleep disturbances in patients affected by cluster headaches. What if the key to studying the physiopathology of such disease could rely on the mechanisms of chronobiology? The purpose of this review is to analyze this link in order to interpret the pathophysiology of cluster headaches and the possible therapeutic implications.

**Keywords:** cluster headache; sleep; chronobiology; chronorisk; circadian rhythm

## 1. Introduction

The cluster headache (CH) belongs to the trigeminal autonomic cephalalgias (TACs), which is a group of primary headaches characterized by unilateral pain accompanied by cranial autonomic symptoms [1]. CH is a very burdensome entity, since its main clinical features are represented by attacks of severe or very severe pain localized in the orbital, supraorbital or temporal region lasting typically from 15 to 180 min that recur several times (typically up to 8) in a day. As already mentioned, the pain usually presents itself concurrently with ipsilateral local autonomic symptoms, such as lacrimation, rhinorrhea, eyelid edema, or sweating, which are typical of TACs (i.e., due to an activation of the trigemino-autonomic reflex) [1,2]. Epidemiological studies show how the lifetime prevalence rate of CH is around 124 per 100,000 inhabitants, affecting up to 0.1% of the population. Moreover, there is a male preponderance (male to female ratio being approximately 2.5:1), while onset age is around 30 years old [3,4]. Furthermore, such disease is responsible for an important reduction in patients' quality of life and many other health-related burdens. For instance, the mean annual direct costs in 2022 amounted to €9158 and €2763 for chronic and episodic patients, respectively. In addition, there is even an associated loss of productivity due to absenteeism with a consequent high indirect cost of €11,809/year/patient in the chronic population and €3558/year/patient in the episodic population [5].

One of the main clinical characteristics of CH is represented by its peculiar temporal presentation: headache episodes recur in variable lifetime periods, namely "clusters" (or bouts), which are separated by pain-free periods (i.e., out-of-bout periods). Depending on the duration of bouts and out-of-bout periods, the International Classification of Headache Disorders, 3rd edition (ICHD-3) distinguishes: "episodic" (i.e., bouts lasting from 7 days to 1 year, separated by out-of-bout periods lasting at least 3 months) and "chronic" (i.e., bouts lasting 1 year or longer without remission, or with out-of-bout periods lasting less than 3 months) CH [1]. In many patients, bouts tend to occur always in the same periods of the year, and during a bout, attacks tend to arise every day at the same time, following a circadian cadence. The circadian and annual periodicity in CH supports an essential link with chronobiology, which is the study of biological rhythm. More in detail, attacks occur mostly at night and follow strongly predictive rhythms throughout the year. This enigmatic and mysterious link with chronobiology could represent an indispensable element in the pathophysiology of CH and a possible key to manage its therapy. Evidence shows how the hypothalamus is implicated in the attacks, and it is not a coincidence that it acts even as a regulatory center of the circadian rhythm [6].

The purpose of this review is to study the link between sleep and CH with particular focus on the underlying chronobiology and the possible therapeutic implications. The most recent full articles written in the English language and indexed in PubMed (including MEDLINE, PubMed Central, and NCBI Bookshelf databases) were included in the present narrative review.

## 2. Chronobiology in Cluster Headache

Chronobiology is the study of biological aspects of human rhythm, and it is strongly linked to CH. In about 82% of CH patients, attacks occur at the same time with a clock-like regularity [7]. Even if in a more variable manner, seasonality is often reported, especially in episodic CH [6].

### 2.1. Diurnal Rhythmicity

Diurnal rhythmicity could rely on an involvement of the biological clock, while seasonal rhythmicity could depend on sunlight exposure. The most important biological rhythm is the one correlated to the light-to-dark cycle; it lasts about 24 h and involves the hypothalamic suprachiasmatic nucleus (SCN), which ensures the rhythm of the whole organism through the melatonin, which is released by the pineal gland [8]. A molecular circuit based on a delayed transcriptional/translational feedback loop (TTFL) could synchronize central and peripheral oscillators and vice versa. However, today, this mechanism is only partially understood, and it is theorized that neural and humoral signals of the SCN together with environmental inputs (e.g., daily fasting–feeding cycles, temperature cycles) and the cellular redox state may represent essential entrainment cues for the peripheral clocks [8]. Circadian rhythmicity in CH refers to the specific timing of attacks during a day and may be influenced even by cultural habits. Most attacks occur during the night or during the early morning, but slight differences may be found in the literature among different populations. For instance, in Northern European patients, attacks tend to occur with two peaks (i.e., between 21.00 and 23.00 and between 04.00 and 10.00—in conditions of relaxation) [9], while in the U.S. ones, the most reported time is 02.00 [7]. Differently, in Italian CH patients, attacks tend to occur at 14.00–15.00 or 21.00–22.00, with such difference perhaps due to local habits since Italian workers usually relax after lunch and dinner [10,11]. On such bases, some authors suggested that it could be possible to modulate the attacks onset by modulating the biorhythm, for instance by administering evening melatonin in episodic CH patients [12]. Furthermore, diurnal rhythmicity may even be influenced by sex differences, as men report attacks between 19.00 and 24.00 more often than women [13]. In addition, women seem to display a phase-shift with a trend toward having attacks one hour later than men, which is probably due to the functional differences of the

hypothalamus [14]. Nevertheless, evidence is still controversial, since other authors described how women show a greater probability to present attacks at night than men [15]. Moreover, another interesting point is that in the early stages of the disease, patients tend to complain of nocturnal attacks, while as the disease progresses, attacks tend to be diurnal [16].

Recently, the concept of "chronorisk" has been introduced to define the probability to present an attack based on the time of the day [17], with interindividual and intraindividual variability [18]. Chronorisk follows a Gaussian distribution with a circadian rhythmicity (cycle of 24 h) in episodic CH and an ultradian rhythmicity (cycle of more than 1 h and less of 24 h) in the chronic subtype [17].

## 2.2. Seasonality

Seasonality (in CH, the circannual presentation of bouts) is a phenomenon probably related to sunlight exposure, which is reported in about half of the patients. It is more easily recognized in episodic, but it is still present even in chronic CH [19]. Evidence shows how the annual recurrence of bouts in episodic patients may occur around 7–10 days after winter or summer solstices (January and July) [20] or in months of the year when the time around-the-clock changes, corresponding to fall and spring [7,13,21]. Following these premises, bouts are demonstrated to be more frequent when days are shorter (i.e., fall) or longer (i.e., spring) [7]. Such seasonality may suggest a sort of "seasonal switch", which could make CH patients susceptible to attacks during bouts periods. Since the hypothalamus is strongly implicated in seasonality, functional Magnetic Resonance Imaging studies were performed comparing resting state signals of in-bout (out of attacks) and out-of-bout patients; as a result, a reduced hypothalamic functional connectivity with medial frontal gyrus, precuneus, and cerebellar areas (tonsil, declive and culmen) was described during in-bout periods [22]. Such results stress the importance of hypothalamus in CH pathophysiology, even if they are not entirely understood yet.

## 2.3. Molecular Mechanisms

In familial CH (i.e., when a patient refers another first- or second-degree relative affected by the same disease), the association between chronobiology and headache is stronger, suggesting that a genetic component is present [23]. CH was recently associated with a polymorphism of the Circadian Locomotor Output Cycles Kaput (CLOCK) gene [24]. This gene, together with basic helix–loop–helix ARNT like 1 (BMAL1) one in mammals, is implicated in the model of TTFL, and it represents the biomolecular mechanism of the circadian oscillations and adaptations to the changing daylight by anticipating behaviors. The beginning of true chronobiology can be traced back to 1984 with the identification of the first clock gene period (Per) [25–27] in the Drosophila; the Per protein product PER shows circadian oscillations with a feedback loop on its own gene transcription. PER translates back to the nucleus thanks to the timeless (TIM) protein (i.e., cryptochrome circadian clock 1—CRY—in mammals), which binds the PER protein to enter the nucleus and to stop the Per gene expression [28]. The processes of transcription and translation are delayed by a doubletime (DBT) gene, obtaining an overall PER/TIM (CRY) period oscillation of about 24 h [29]. The positive spontaneous transcription factors of PER and TIM (CRY) are two other gene products CLOCK and BMAL1 in mammals (and their analogous CLK and CYC in the Drosophila); CLOCK and BMAL1 are in turn inhibited by the PER and TIM (CRY) proteins, closing the feedback loop (see Figure 1) [30]. Another gene implicated in CH patients is the one of the core circadian gene Nr1d1 (encoding REV-ERBα, a circadian transcription repressor), which showed a markedly decreased expression [31]. Moreover, RBM3 (encoding RNA binding motif protein 3) regulates alternative polyadenylation and thereby circadian gene expression level and amplitude, and it is the most altered gene in CH [32].

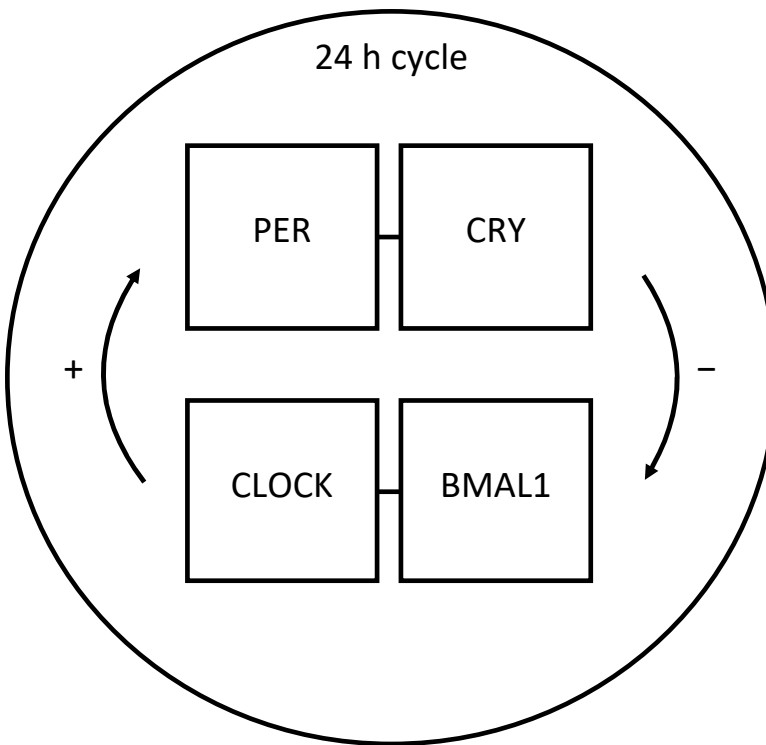

**Figure 1.** Schematic representation of the biomolecular mechanism of the circadian cycle in mammals: the transcriptional translational feedback loop (TTFL). PER and CRY inhibit the transcription of CLOCK and BMAL1, which in turn are responsible for PER and CRY transcription. The overall cycle lasts about 24 h. PER: period gene product; CRY1: cryptochrome circadian clock 1; CLOCK: Circadian Locomotor Output Cycles Kaput; BMAL1: basic helix–loop–helix ARNT like 1.

### 3. Sleep Disorders and CH

The relationship between headache and sleep disorders is complex and often bidirectional, since a headache may be a consequence of sleep disorders or be aggravated by them. For instance, morning headaches are associated with insomnia, loud snoring, sleep-related breathing disorders, nightmares and other sleep disorders [33]. Sleep and headaches find a binomial relation even in the case of CH [34]: as a matter of fact, the subjective sleep quality is reduced both in bouts and in out-of-bout periods. Furthermore, sleep is reported as a CH attack trigger by 80% of patients [35]. Thus, we may wonder if sleep alterations primarily affect CH or vice versa. In the following paragraphs, we will summarize the current evidence exploring the impact of sleep disorders in CH.

#### 3.1. REM Sleep

Originally, the occurrence of CH attacks was associated with REM sleep due to the finding of higher frequency of attacks 90 min after sleep onset, which coincides with the first REM phase of sleep [6,36]. In addition, CH patients awakened at night by an attack are usually able to recall their dreams, which should mean that attacks occur during REM sleep. Overall, studies exploring the sleep phases in CH patients showed: (1) irregular sleep–wake patterns; (2) a decreased total amount of sleep; (3) a decreased proportion of REM sleep; (4) a prolongation of REM latency; and (5) a condition of hypoarousability during REM sleep in bouts [6,37]. Nevertheless, further studies failed to prove an association between CH attacks and REM sleep [38–40]. The evidence about the association between REM sleep and CH attacks is conflicting even if REM sleep may not represent a prerequisite for CH attacks, since they have been observed during other sleep phases, too [40]. Such evidence could support previous hypotheses that the beginning and the end stages of sleep may represent the more vulnerable phases [6,41].

### 3.2. Chronotypes and Sleep Quality

CH attacks often occur during the night, and they usually present a circadian and circannual rhythmicity. Overall, such circannual rhythmicity is more reported by episodic CH patients than chronic ones, while no significant differences are described in circadian rhythmicity between the two subgroups of patients. In addition, patients who complain of diurnal rhythmicity commonly report sleep as a trigger [6]. Furthermore, a troubled sleep, too much sleep, and waking up late have been associated with a higher probability of developing an attack. The link between CH and circadian rhythm has also been studied by evaluating the prevalent chronotypes (i.e., the individual behavioral preference about the time to go to sleep). Patients affected by episodic CH show a late chronotype, whereas chronic CH patients often show an early one [42]. Furthermore, the ability to adapt to sleep deprivation demonstrated to be more in chronic patients compared to episodic ones [42]. Apart from chronotype considerations, sleep quality and other associated sleep disturbances should not be neglected. Indeed, several studies reported a lower sleep quality in patients affected CH compared with controls, even one year after the last attack [6,42]. In addition, chronic insomnia and shift work is common among CH patients as well, and an association between insomnia and longer lasting cluster bouts has been demonstrated [21]. Regarding sex differences in CH, women with chronic CH tend to take longer to fall asleep compared with men, and this may be associated with bad sleep quality and a delayed time to first nightly attack in women [14]. Moreover, a slow recovery or a persistence of sleep disturbances even outside the clusters were demonstrated, suggesting that sleep disturbances are not just secondary to attacks [43]. Thus, the presence of a relationship between CH and sleep seems to be clear, but the causal connection has not been clarified in the literature yet, and it is unknown whether both sleep disturbances and CH may belong to a wider spectrum of chronobiology disorders. Finally, the hypothalamus (implicated in sleep and chronobiology) may play a central role in the pathophysiology of such disorders, since there is evidence that sleep quality may return to normal conditions in CH patients who undergo deep brain stimulation (DBS) of the posterior hypothalamus [44].

### 3.3. Sleep Breathing Disorders

CH shows another connection with sleep if we look at sleep disordered breathing (SDB), particularly regarding obstructive sleep apnea (OSA). Due to the higher prevalence of OSA in patients with CH compared to healthy controls (i.e., from 58% to 80%) and the higher prevalence of CH attacks during the night, the causal relationship between these two conditions has been hypothesized [45–47]. Of interest, OSA severity is higher during REM sleep, which is the sleep phase putatively associated with an increased risk of CH attacks. Thus, we may wonder whether sleep apnea may trigger CH attacks or if it represents an associated and concomitant factor with CH. Studies conducted on CH patients demonstrate how, during bouts, such patients present an increased prevalence of OSA syndrome and a higher respiratory distress index than healthy controls [48]. The relationship between sleep apnea and CH is conflicting: probably, sleep apnea is not itself able to trigger nocturnal CH attacks, but both disturbances are resulting from a central hypothalamic dysfunction. Indeed, the anterior hypothalamus has been found to be involved in sleep apnea, since the preoptic area promotes sleep and the reduction in muscle activation [49]. Thus, sleep apnea and CH may be two parallel consequences of a hypothalamus dysfunction rather than being causative for each other [50]. According to ICHD-3 classification, the distinction between CH with sleep apnea and headache secondary sleep apnea should be looked at carefully to avoid misdiagnoses [1].

### 3.4. The Role of Hypothalamus

Given the peculiar chronobiological nature of the CH attacks, it has been hypothesized that the hypothalamus plays a fundamental role in the periodic nature of CH and its relationship with sleep. Indeed, the anterior hypothalamus hosts the SCN, which represents the biological clock, while the posterior hypothalamus hosts the hypocretins secreting neurons,

which are involved in pain processing, sleep, and arousal [51–53]. Furthermore, sex differences in CH chronobiology may be related with the sexual dimorphism of hypothalamic nuclei, which have different functions in men and women. Such differences could be influenced by sexual hormones, but data in this area are still limited, with the association among hormones, CH, chronobiology, and clinical characteristics that remains just possible [14,54]. In particular, the diminished influence of SCN on diencephalic and brainstem circuits in chronic CH has been thought of as responsible for these sex differences [40].

Based on the hypothesis of SCN involvement in CH pathogenesis, the polymorphisms of clock gene PERIOD3 (implicated in circadian misalignment) were investigated, but unfortunately without any conclusive finding [55]. On the other hand, a higher activation of the ipsilateral posterior hypothalamus has been found in episodic CH patients during an attack, suggesting an involvement of this area in CH pathogenesis [56]. In this view, through the hypocretinergic projections, the hypothalamus plays a role in the central pain processing network via the bidirectional connection with trigeminal nuclei, exerting a descending control over the trigeminal cervical complex [57]. Particularly, hypocretin-1 and 2 show pro- and antinociceptive effects, respectively. Some authors found an association between a polymorphism in the hypocretin-2 receptor and CH, while others did not find any association, suggesting that the genetic factors are heterogeneous [58,59]. Moreover, studies on CSF hypocretin-1 levels in patients with episodic CH provide controversial results, being normal or low [60,61]. Studies on rats demonstrated that there is a mutual connection between the hypothalamus and the trigeminal nuclei via the trigeminohypothalamic tract for both nociceptive and non-nociceptive signals [62]. Such connections may be at the origin of the believed influence of hypothalamic structures on the trigemino-autonomic reflex, which is one of the main actors implicated in CH symptoms [63]. Finally, the DBS of posterior hypothalamus in patients with CH led to nocturnal CH attacks abortion, improvement of sleep quality and sleep efficiency, increasing of slow-wave sleep and reduction in sleep fragmentation as well as reduction in sympathoexcitatory firing [64,65]. The role of hypothalamus in CH pathogenesis may be resumed in three points: (1) altered chronobiology (e.g., diminished arousal, altered REM-sleep, sleep fragmentation); (2) altered descending nociceptive control; and (3) altered autonomic system. However, both anterior and posterior hypothalamus are involved in CH pathogenesis together with REM sleep and sleep apnea representing another connection tract between CH and sleep.

Altogether, the presented evidence suggests that sleep disorders are not a consequence of acute CH attacks, but CH is a chronobiology disorder involving CNS structures, and sleep disorders may represent an epiphenomenon of this involvement. Thus, larger longitudinal studies are needed in the future to better assess the relationship between the type of sleep disorder and CH focusing on hypothalamic functions in order to achieve an effective treatment for these patients.

## 4. Melatonin and Autonomic Dysfunction

Melatonin is a hormone produced by the pineal gland under the influence of the hypothalamic SCN and reflecting circadian fluctuations of the organism. The evidence of a role for melatonin in CH pathogenesis is scarce, while data demonstrate that its nocturnal plasmatic levels are lower during clusters compared to controls and remission periods [66]. Such a finding is thought to be actually dependent on the CH hypothalamic dysfunction rather than being considered pathogenetic. Moreover, data in the literature point even to a possible role of melatonin in the induction of peripheral vasodilation, but whether this occurs through its direct action on blood vessels or by means of a modulation of sympathetic activity remains unclear [67].

Although CH is mainly characterized by local autonomic manifestations, there are several studies in the literature that tried to reveal a possible systemic involvement of the sympathetic or parasympathetic nervous system [68,69]. Neurophysiological studies demonstrated how around 15% of patients may present orthostatic hypotension during bouts, as well as alterations in the forehead sympathetic skin reflex, compared to out-

of-bouts periods or healthy subjects [70]. Thus, a local involvement of the autonomic seems to be prevalent, even if other studies with frequency domain heart rate variability analysis report a blunted shift from parasympathetic predominance to the sympathetic one during a head-up tilt table test that was interpreted as a central dysregulation due to hypothalamic involvement [71]. Furthermore, even baroreflex sensitivity was shown to be altered (reduced) in CH, believing it could have been an epiphenomenon due to the central autonomic dysregulation [72]. Other authors confirmed also the presence of alterations in parasympathetic cardiovascular responses in CH patients [73], but further ones did not find any cardiovascular difference between in bouts and out-of-bout patients [74]. To date, a systemic involvement of the autonomic system in CH is suggested, but data are still controversial in the literature, and further studies on larger populations are required.

## 5. Possible (Sleep Related) Therapeutic Implications

Many of the drugs commonly used as prophylaxis in CH also influence the sleep pattern: lithium, which is efficaciously used to treat also other disorders that show a periodic pattern (e.g., bipolar disturb, cyclic migraine, hypnic headache), alters the circadian rhythm and dampens REM sleep [75]; anticonvulsants such as topiramate, pregabalin and gabapentin increase REM sleep duration and reduce sleep latency [40]; verapamil showed circadian effects both on cellular and behavioral levels in mice [76]. On the other hand, since there is a close relationship between sleep and CH, it is reasonable to think that using therapeutic strategies to modulate the former would influence the latter. For instance, there is some evidence that oral melatonin may produce a significant reduction in headache frequency and the lowering of analgesic consumption [12], even if such a finding was not confirmed by further studies [77]. Another sleep approach therapy may be found with drugs acting as regulators of the circadian rhythm, such as Ramelteon, which is a selective melatonin MT1/MT2 receptor agonist used in insomnia therapy; there is a case report about a CH patient with associated insomnia who significantly benefited from Ramelteon use, showing a significant reduction in headache frequency [78]. Sodium oxybate (SO), the sodium salt of γ-hydroxybutyric acid (GHB), is currently approved for the treatment of narcolepsy with cataplexy, improving slow-wave sleep and sleep quality in general. Even here, there is evidence that it may lead to a very positive effect in preventing headache attacks (up to 90%) and in reducing their intensity (up to 50%) in chronic CH [79]. In the literature, the relationship between sleep apnea in patients with OSA and CH has been long established [50]. Despite there being few case reports of CH patients improving with OSA treatment [80], evidence is still weak and results are controversial. Thus, OSA and CH seem to represent parallel phenomena influenced by a common central cause, lying in a hypothalamic dysfunction [50]. Furthermore, functional studies evidenced the central role of the posterior hypothalamus in triggering CH attacks and revealed how this structure is also implicated in pain modulation through its connection with brainstem [81]. In the same direction, the DBS of the posterior hypothalamus may represent an effective treatment for drug-resistant CH. While its mechanism of action remains still unclear, there are studies reporting a general improvement in patients' increased total sleep time, sleep efficiency and the amount of slow-wave sleep stages [64].

Although some of the above-mentioned sleep-related therapeutic strategies are already available, further research with larger populations and double-blind randomized trials is needed to improve the clinical management of CH in the near future. The development of a specific chronotherapy composed of pharmacological and non-pharmacological interventions could represent a successful way to approach such intricate disorder.

Finally, Galcanezumab, a monoclonal antibody targeting the calcitonin gene-related peptide (CGRP), has been recently approved by the FDA for episodic CH preventive treatment based on the results of clinical trials [82]. CGRP's role in the pathophysiology of CH is still not completely understood; lower blood levels of CGRP were found in patients with episodic CH compared to chronic CH, while in migraines, CGRP levels are proportional to headache frequency [83]. Furthermore, CGRP is involved in complex

pathways that converge on thalamic trigeminovascular neurons that might modulate pain sensitivity in primary headaches in response to various conditions, including sleep deprivation [84]. Regarding sleep influence, Galcanezumab improved sleep quality in patients suffering from migraine [85]; whether the effectiveness of Galcanezumab on CH could be explained by sleep quality improvement is worth exploring through more specific investigations.

## 6. Conclusions

The relationship between sleep and cluster headache is complex and mysterious and is based on anatomical, chronobiological, and genetic links. The rhythm of the attacks could depend on a common anatomical locus that can be found in the hypothalamus. Melatonin secretion and related autonomic disorders are implicated in the chronobiology of the CH. Based on the connection among sleep, circadian rhythm and CH, we look toward an understanding of the pathophysiology of this headache in order to realize more and more personalized therapies. However, further studies are needed on this topic.

**Author Contributions:** Conceptualization, L.P. and A.T.; methodology, A.T. and V.D.S.; resources, A.T.; writing—original draft preparation, A.T., L.P., P.A., L.V., S.M., A.G., S.F., A.P. and S.I.; writing—review and editing, A.T. and L.P.; visualization, P.A. and A.G.; supervision, F.B. and C.C.; project administration, F.B. All authors have read and agreed to the published version of the manuscript.

**Funding:** This research received no external funding.

**Institutional Review Board Statement:** Not applicable.

**Informed Consent Statement:** Not applicable.

**Data Availability Statement:** Not applicable.

**Conflicts of Interest:** The authors declare no conflict of interest.

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
