# Peer review of "Sleep and Chronobiology as a Key to Understand Cluster Headache"

_2035-8377, doi:10.3390/neurolint15010029_

Round 1

Reviewer 1 Report

The authors present a review article focusing on the association between cluster headache and sleep/chronobiology. This is certainly an interesting topic that has received reasonable attention to date with several existing reviews available. 

Overall the review would benefit from some proof reading as often the wording is not ideal or sentences appear truncated. This occasional grammatical lapse is further highlighted by regions with a somewhat bullet style discussion. Often the authors discuss study by study or point by point and this disrupts the narrative giving the article a reduced feel of being an expert statement.

I have included some suggestions below as examples of areas for attention:

1.    When discussing the SCN (line 77) the authors fail to acknowledge that normal physiological function relies on a coordinated interaction between the master SCN clock and local “body” clocks present in the majority of tissues. This detail should be added as it may be misalignment of these clocks that underlies this and other disorders.

2.    When discussing attack timing, the authors jump from using a 12-hr to a 24-hr clock and this is confusing and likely reflects the time stated in the original article. In this case I would suggest using the 24-hr clock throughout. Following from this, on line 93 the authors state attacks between 19 and 24 more often… It needs to be clear that this is a time (19.00 – 24.00) as at first the ambiguity of the style had me thinking age.

3.    The claim of a functional hypothalamic difference underlying the sex-specific difference in CH timing is a somewhat tenuous claim with little direct evidence.

4.    When discussing bouts, it is important to highlight the magnitude of the change from out of bout, to in bout. Out of bout, patients can’t be triggered but in bout they can, so this suggests a critical seasonal switch, which merits mentioning.

5.    Consider splitting circadian and seasonal discussions with a more obvious sub-heading as these are very different aspects.

6.    Around line 207 the authors introduce a point about OSA and then casually state that this is confirmed by the fact that oxygen is able to suppress CH attacks. Then they introduce the fact that positive airway pressure does not impact this, disproving this point. Overall I would rather see a more balanced discussion that states that while an association has been proposed the evidence is contradictory… Currently it appears as a fact and then this is undermined. This happens often enough I the manuscript that the non-expert reader may get confused to the overall evidence on a point. Further, no references are given to support the role of oxygen.

7.    Near line 240 the authors discuss hypocretinergic projections, but use a review to cite this rather than the original research. Original research sources should always be used where possible.

8.    An example of the wording issue occurs on line 248. ‘If the role of the hypothalamus in CH pathogenesis is almost likely’. What does almost likely mean, it is a rather diffuse wording for an expert review.

9.    I would like to see some wider discussion around the role of the hypothalamus in the regulation of the cranial autonomic pathway, e.g. via the PVN-SSN that many consider may alter the activation threshold for the trigeminal-autonomic reflex. The hypothalamus is very complex and the authors only scratch the surface of its potential involvement in the current text.

Author Response

The authors present a review article focusing on the association between cluster headache and sleep/chronobiology. This is certainly an interesting topic that has received reasonable attention to date with several existing reviews available. 

Overall, the review would benefit from some proof reading as often the wording is not ideal or sentences appear truncated. This occasional grammatical lapse is further highlighted by regions with a somewhat bullet style discussion. Often the authors discuss study by study or point by point and this disrupts the narrative giving the article a reduced feel of being an expert statement.

Authors thank the Reviewer for taking the time to review the manuscript and for the timely revision. The Reviewer’s suggestions were very helpful in improving the overall quality of the manuscript. We answered point by point to the Reviewer’s suggestions and performed an extensive revision of the manuscript, improving English grammar and unifying narrative style. Authors hope to find Reviewer’s approval with the current version of the manuscript.

I have included some suggestions below as examples of areas for attention:

  1. When discussing the SCN (line 77) the authors fail to acknowledge that normal physiological function relies on a coordinated interaction between the master SCN clock and local “body” clocks present in the majority of tissues. This detail should be added as it may be misalignment of these clocks that underlies this and other disorders.

Authors thanks the Reviewer for the revision. We added a short description of the suggested interaction between internal SCN clock and the peripheral clocks: “However, today, this mechanism is only partially understood, and it is theorized that neural and humoral signals of SCN together with environmental inputs (e.g., daily fasting-feeding cycles, temperature cycles) and the cellular redox state represent essential entrainment cues for the peripheral clocks [7].”

  1. When discussing attack timing, the authors jump from using a 12-hr to a 24-hr clock and this is confusing and likely reflects the time stated in the original article. In this case I would suggest using the 24-hr clock throughout. Following from this, on line 93 the authors state attacks between 19 and 24 more often… It needs to be clear that this is a time (19.00 – 24.00) as at first the ambiguity of the style had me thinking age.

Thanks for the suggestion. To avoid misunderstandings, we standardized the times all according to 24hr clock (e.g., 19.00-24.00).

  1. The claim of a functional hypothalamic difference underlying the sex-specific difference in CH timing is a somewhat tenuous claim with little direct evidence.

Authors agree with the Reviewer on the scarce evidence, but we added few hypotheses from the literature: “Furthermore, sex differences in CH chronobiology may be related with the sexual dimorphism of hypothalamic nuclei, which have different functions in men and women. Such difference could be influenced by sexual hormones, but data in this area are still limited, with the association among hormones, CH, chronobiology, and clinical characteristics that remains just possible [13,53]. In particular, the diminished influence of SCN on diencephalic and brainstem circuits in chronic CH has been thought as responsible for these sex differences [40]”

  1. When discussing bouts, it is important to highlight the magnitude of the change from out of bout, to in bout. Out of bout, patients can’t be triggered but in bout they can, so this suggests a critical seasonal switch, which merits mentioning.

We followed the Reviewer’s suggestion and mentioned in-bouts and out-of-bouts period both in the introduction and in the chronobiology paragraphs, stressing hypothalamus role.

  1. Consider splitting circadian and seasonal discussions with a more obvious sub-heading as these are very different aspects.

Done, we believe paragraph 2 is now clearer.

  1. Around line 207 the authors introduce a point about OSA and then casually state that this is confirmed by the fact that oxygen is able to suppress CH attacks. Then they introduce the fact that positive airway pressure does not impact this, disproving this point. Overall I would rather see a more balanced discussion that states that while an association has been proposed the evidence is contradictory… Currently it appears as a fact and then this is undermined. This happens often enough I the manuscript that the non-expert reader may get confused to the overall evidence on a point. Further, no references are given to support the role of oxygen.

The cited expression was removed and authors added relevant evidence based information on the subject.

  1. Near line 240 the authors discuss hypocretinergic projections, but use a review to cite this rather than the original research. Original research sources should always be used where possible.

Thanks for the observation. We have appropriately reviewed the reference and literature

  1. An example of the wording issue occurs on line 248. ‘If the role of the hypothalamus in CH pathogenesis is almost likely’. What does almost likely mean, it is a rather diffuse wording for an expert review

Checked.

  1. I would like to see some wider discussion around the role of the hypothalamus in the regulation of the cranial autonomic pathway, e.g. via the PVN-SSN that many consider may alter the activation threshold for the trigeminal-autonomic reflex. The hypothalamus is very complex and the authors only scratch the surface of its potential involvement in the current text.

Thanks for the observation, we added information about the trigeminohypothalamic tract and about its importance in hypothalamic influence on trigeminovascular reflex.

Reviewer 2 Report

The review entitled "Sleep and chronobiology as a key to understand cluster headache" aimed to analyze the link between the physiopathology of cluster headache and the mechanisms of chronobiology, in order to interpret the pathophysiology of cluster headache and its possible therapeutic implications. The review is well organized but there are a few aspects that could be improved in order to reflect a more realistic and complex perspective upon this healthcare issue.

The presence of the 2 figures is absolutely elusive and I suggest the authors to renounce to them, as the lack of scientific meaning and academic quality rather decreases the interest of readers at a first look.

The review should contain some demographic concludent data on the populations mostly exposed to the risk of cluster headache, the incidence of this pathology in the general population and the extent of the burden represented by this disease on the quality of life of people subjected to it.

There should be included a schematic detail representation of the biomolecular mechanism of the circadian oscillation, that is described in paragraph 123-143.

A discussion regarding the trends in chronotherapy and the use of novel pharmacologic targets and molecules should also increase the scientific soundness of this review.

Author Response

Reviewer #2 answers:

The review entitled "Sleep and chronobiology as a key to understand cluster headache" aimed to analyze the link between the physiopathology of cluster headache and the mechanisms of chronobiology, in order to interpret the pathophysiology of cluster headache and its possible therapeutic implications. The review is well organized but there are a few aspects that could be improved in order to reflect a more realistic and complex perspective upon this healthcare issue.

Authors thank the Reviewer for taking the time to review the manuscript, the suggestions were very helpful in improving the overall quality of the manuscript. We answered point by point to the Reviewer’s suggestions and hope to find his/her approval with the current version of the manuscript.

The presence of the 2 figures is absolutely elusive and I suggest the authors to renounce to them, as the lack of scientific meaning and academic quality rather decreases the interest of readers at a first look.

Thanks for the observation. As suggested, authors removed the figures.

The review should contain some demographic concludent data on the populations mostly exposed to the risk of cluster headache, the incidence of this pathology in the general population and the extent of the burden represented by this disease on the quality of life of people subjected to it.

Thanks for the suggestion, we added the incidence and the burden of cluster headache.

There should be included a schematic detail representation of the biomolecular mechanism of the circadian oscillation, that is described in paragraph 123-143.

We added a figure (figure 1) with the schematic representation requested. Authors believe that this paragraph now results clearer thanks to the Reviewer’s advice.

A discussion regarding the trends in chronotherapy and the use of novel pharmacologic targets and molecules should also increase the scientific soundness of this review.

Unfortunately, there is no report of any chronotherapeutical intervention in CH. Reviewer’s suggestion pointed an interesting subject that may be the object of future research. Chronotherapy has now been mentioned in the manuscript, as well as the possible role of novel anti calcitonin gene-related peptide (CGRP) monoclonal antibodies on sleep modulation and CH prevention.

Round 2

Reviewer 1 Report

While the authors have made a reasonable attempt to adress key points some small (editing mostly) points remain.

Introduction, page 1 line 44-45. It is the trigeminal-autonomic reflex not the trigeminovascular reflex.

The male to female ratio has decreased somewhat with better education and correct diagnosis in females and is now considered to be towards the lower end of the estimates given early on page 2.

The issue of timing (24hr clock) is mostly resolved except for in one place. On page 2 near the bottom the authors still use 2.00 (is this am or pm), and using the 24hr clock correctly would remove this ambiguity.

Overall a further text edit would be beneficial to further reduce the bullet style discussion of paper by paper. The role of such a review is to assimilate the available literature and explain it in a clear narrative and as such the current study by study approach somewhat detracts from the expert feel of the review. One example of this is the highlighted diurnal section on page 2-3.

Other areas that require grammatical editing remain. One example is page 4, line 169-171 ending in “both in bouts and in out-of-bout periods them”. Them should be removed..

Author Response

While the authors have made a reasonable attempt to adress key points some small (editing mostly) points remain.

Authors thank the reviewer for the time spent on reviewing the manuscript again. We believe that now the overall quality and style improved a lot. English style was carefully improved by the two main authors and all the minor errors were corrected.

Introduction, page 1 line 44-45. It is the trigeminal-autonomic reflex not the trigeminovascular reflex.

Checked.

The male to female ratio has decreased somewhat with better education and correct diagnosis in females and is now considered to be towards the lower end of the estimates given early on page 2.

Thanks for the correction, we remodulated following Reviewer’s suggestion.

The issue of timing (24hr clock) is mostly resolved except for in one place. On page 2 near the bottom the authors still use 2.00 (is this am or pm), and using the 24hr clock correctly would remove this ambiguity.

Corrected.

Overall a further text edit would be beneficial to further reduce the bullet style discussion of paper by paper. The role of such a review is to assimilate the available literature and explain it in a clear narrative and as such the current study by study approach somewhat detracts from the expert feel of the review. One example of this is the highlighted diurnal section on page 2-3.

Authors performed an extensive revision of manuscript style, using a more narrative one as suggested. We believe that now the paper is clearer and easier to read.

Other areas that require grammatical editing remain. One example is page 4, line 169-171 ending in “both in bouts and in out-of-bout periods them”. Them should be removed..

We corrected the error and checked the whole manuscript for further ones.

Reviewer 2 Report

The present form of the article is substantially improved and is further accepted for publication.

Author Response

Authors thank the reviewer for the time spent on the manuscript and for the previous suggestions that increased the quality and the soundness of the paper very much.